# Online Evaluation Method for Low Frequency Oscillation Stability in a Power System Based on Improved XGboost

**Wenping Hu [1],\*, Jifeng Liang [1], Yitao Jin [2], Fuzhang Wu [2], Xiaowei Wang [1] and Ersong Chen [1]**

[1]   State Grid Hebei Electric Power Co., Ltd., Technology Research Institute, Shijiazhuang 050021, China; dyy_liangjf@he.sgcc.com.cn (J.L.); dyy_wangxw@he.sgcc.com.cn (X.W.); dyy_chenes@he.sgcc.com.cn (E.C.)
[2]   School of Electrical Engineering, Wuhan University, Wuhan 430072, China; 2017282070208@whu.edu.cn (Y.J.); 2014302580081@whu.edu.cn (F.W.)
\*   Correspondence: dyy_huwp@he.sgcc.com.cn; Tel.: +86-0311-6667-3203

**Abstract:** Low frequency oscillation in an interconnected power system is becoming an increasingly serious problem. It is of great practical significance to make online evaluation of actual power grid's stability. To evaluate the stability of the power system quickly and accurately, a low frequency oscillation stability evaluation method based on an improved XGboost algorithm and power system random response data is proposed in this paper. Firstly, the original input feature set describing the dynamic characteristics of the power system is established by analyzing the substance of low frequency oscillation. Taking the random response data of power system including the disturbance end time feature and the dynamic feature of power system as the input sample set, the wavelet threshold is applied to improve its effectiveness. Secondly, using the eigenvalue analysis method, different damping ratios are selected as threshold values to judge the stability of the system low-frequency oscillation. Then, the supervised training with improved XGboost algorithm is performed on the characteristics of stability. On this basis, the training model is obtained and applied to online low frequency oscillation stability evaluation of a power system. Finally, the simulation results of the eight-machine 36-node test system and Hebei southern power grid show that the proposed low frequency oscillation online evaluation method has the features of high evaluation accuracy, fast evaluation speed, low error rate of unstable sample evaluation, and strong anti-noise ability.

**Keywords:** random response data; low frequency oscillation stability; online evaluation; improved XGboost algorithm

## 1. Introduction

With the development of ultra-high voltage (UHV) transmission technology and flexible alternating-current (AC) transmission technology, modern power systems have entered the era of large units, UHV, super large scale, long distance, alternating-current and direct-current (AC-DC) hybrid transmission. The interconnection of regional power grids is becoming more and more compact, and the scale of the system is increasingly complex. As the grid operates in a variety of ways and the dynamic characteristics are more complex, the occurrence of low-frequency oscillations will have a serious impact on the grid. It is important to evaluate the stability of low-frequency oscillations online.

Low frequency oscillation which is closely related to small signal stability is usually attributed to small signal stability analysis. The small signal stability analysis of the power system includes frequency domain analysis, eigenvalue analysis and time domain analysis [1–4], but these methods

don't consider the actual uncertainties, and it is difficult to fully reflect the stability level of low-frequency oscillations in actual systems. Therefore, the probabilistic analysis method is introduced. And the statistical probability index of the small signal stability is established by considering the random variables such as state and force variation, load fluctuation, and line parameter variation under various working conditions [5,6]. In literature [7], a small signal stability frequency estimation method is proposed by introducing the Monte Carlo method to the random variables such as load level and form, generator state, and network topology parameters. However, the probability model of random variables is relatively simple, so that the evaluation results cannot accurately reflect the actual situation of the grid. In literature [8,9], the problem is solved. Complex systems require a large amount of computation and long simulation time, so it is necessary to further study more effective methods for evaluating the low-frequency oscillation stability. Based on the eigenvalue analysis method and risk assessment method, considering the probability safety and instability of the system, the literature [10] proposed a method to quickly evaluate the real-time risk of small-scale power grid, but did not consider the uncertainty of the grid. Literature [11] studies the probability distribution of system vibration modal damping based on the deterministic small-signal safety analysis when considering uncertainties. However, how to evaluate low frequency oscillations stability had not been studied. Considering the seriousness of system instability, the literature [12] proposed a risk-based probabilistic small-signal safety analysis method, and it quantified risk through matrix and continuous function. This method takes into account uncertainties of the power system. A nomogram method based on the analysis of oscillation damping factors is used for small-signal security assessment of power systems to increase accuracy [13], but these methods still belong to offline evaluation and the results of estimates are inaccurate. This literature [14] describes that the small signal stability assessment with phasor measurement can be applied online, but the timeliness of judgment is poor. Then a new data-driven methodology to detection of low-frequency oscillations is proposed in [15] and literature [16] presents a risk-based probabilistic small-disturbance security analysis (PSSA) methodology for use with power systems with uncertainties, these methods are rapidity but the accuracy needs to be improved.

During the daily operation of the power system, there are persistent small signals of random nature such as load variation, tap changer of transformer, and so on, which bring some random disturbance to the system. The random response data obtained by measurement is externally characterized as a random, noise-like random response data [17]. This kind of data is not only rich and easy to obtain, but also it contains a large number of electromechanical oscillation characteristics related to actual working conditions, which implies the uncertainties of the actual grid during operation. The low frequency oscillation stability evaluation method based on random response data has received extensive attention. Literature [18,19] use frequency domain decomposition and the total least squares-rotation invariant technique to extract vibration information from random response data. The stochastic subspace identification (SSI) method has become a common method for low-frequency oscillation identification [20]. Because its model order is simple, it has high adaptability to systems with large data volume and complex dynamic processes. Forgetting factor is introduced into the original recursive stochastic subspace identification (RSSI) algorithm in [21], which improves the calculation speed of system model parameters. However, the selection of genetic factors is very importance and it is difficult to find suitable genetic factors in practice. In literature [22], a new Bayesian method for the measurement based analysis of electromechanical modes is proposed, which can accurately identity. However, the power system in actual operation is often affected by various small disturbances, and the above methods have a low recognition speed and cannot meet the requirements of real-time evaluation.

With the rapid development of artificial intelligence (AI), the use of data-driven methods to study grid security issues have become a new approach. The AI technology is applied to the analysis of low-frequency oscillation stability for the first time. A neural network-based eigenvalue prediction method for power system critical stability model is proposed in [23]. Although it has high accuracy, it is offline evaluation. Therefore, the use of artificial intelligence technology to solve the problem of low frequency oscillation stability is a new direction. XGboost (Extreme gradient boosting) is a large-scale

parallel learning algorithm which uses different processing methods to learn how to handle missing values when different nodes encounter missing values. Moreover, it has the advantages of low input data requirement, automatic variable selection, and low computational complexity [24]. And it has been applied in the field of wind turbine fault detection [25]. Literature [25–27] show that the XGboost classifier not only has faster prediction speed than the other classifiers such as support vector machine (SVM) and deep belief network (DBN), but it also has higher prediction accuracy.

The main contributions of this paper is to propose a machine learning method to evaluate the low-frequency oscillation stability of the power system timely and accurately considering the random response data containing the uncertainties of the power grid. Firstly, the original input feature set of the evaluation system is established to ensure the efficiency of the evaluation by analyzing the effects of generator electromechanical model, excitation system, and PSS on low frequency oscillation. Secondly, the data mining method and the improved XGboost machine learning method are applied to analyze the random response data, and then the supervised training is conducted to obtain the training model that describes the relationship between feature set and low-frequency oscillation stability. Finally, the model is applied to online evaluation of low frequency oscillation stability.

The rest of this paper is organized as follows: analyze the essence of low frequency oscillations and establish the original input characteristics of low frequency oscillations in Section 2. Section 3 introduces the principle of XGboost and improves the XGboost algorithm. XGboost classifies the random response data after wavelet threshold de-noising and z-score normalization. An online evaluation model for low frequency oscillations is proposed and a model performance evaluation index is established in Section 4. Simulations and analysis are shown in Section 5. Finally, conclusions are drawn in Section 6.

## 2. The Construction of the Original Input Feature

The main factors affecting the stability of low-frequency oscillation in power system include the initial operation state, the tightness of the components in the transmission system and the features of various control devices. And the specific disturbance values and forms are independent of the low frequency oscillation stability. Therefore, the low frequency oscillation stability can be judged by calculating the damping ratio of the system oscillation mode. In this paper, different damping ratios are chosen as the threshold of low frequency oscillation stability damping ratio. According to the threshold of damping ratio, the low frequency oscillation stability is divided into three categories: (1) Negative damping; (2) Weak damping; and (3) Strong damping.

By analyzing the essence of low frequency oscillation stability, a set of original input features for online evaluation of low frequency oscillation stability is constructed.

The third order generator model is adopted and the differential equation of generator is incremented as ($P_m = 0$):

$$\begin{cases} T'_{d0}p\Delta E'_q = \Delta E_f - \Delta E_q \\ Mp\Delta\omega = -\Delta P_e - D\Delta\omega \\ p\Delta\delta = \Delta\omega \end{cases} \tag{1}$$

In Formula (1), $\Delta E_f$ is the change of output excitation voltage of the excitation system; $\Delta E'_q$ is the change of transient potential of the $q$ axis; $\Delta\omega\Delta P_e$ and $\Delta\delta$ are electromagnetic power, rotor angular velocity and rotor angle respectively; $M$ and $D$ are inertia time constant and self-damping coefficient.

Therefore, analysis Formula (2) shows that $\Delta\omega$, $\Delta P_e$, and $\Delta\delta$ describe the change features of the generator when the power system is subjected to small disturbance. The following feature sets can be selected: the change values of the electromagnetic power per unit time, that is the electromagnetic accelerate power (maximum, minimum, and average); the change in angle per unit time, that is the angular velocity; the velocity (the difference between the maximum and the minimum of the angular velocity); the change in angular velocity per unit time, that is the angular acceleration (the difference between the maximum and the minimum of the angular acceleration).

Transfer function of excitation system, set ($U_{ref} = const$):

$$\frac{\Delta E_f}{-\Delta U_t} = \frac{K_E}{1 + T_E p} = G_E(p). \tag{2}$$

Increments the excitation system to:

$$T_E p \Delta E_f = -\Delta E_f - K_E \Delta U_t. \tag{3}$$

In Formula (2), $\Delta U_t$ is the change value of generator terminal voltage.

Therefore, analysis Formula (3) shows that $\Delta E_f$ describes the change features of the excitation system when the power system is subjected to small disturbance, and the following feature sets can be selected—the change value of the excitation voltage per unit time (the maximum value, the minimum value, and the mean value of the change value).

Due to the large electromagnetic inertia of the excitation system, the negative damping caused by the regulator under certain conditions (high load level and weak connection) will have a negative impact on the dynamic stability of the power system and it causes low frequency oscillation. The principle of PSS is as follows: when the system is subjected to low frequency oscillation after small disturbance, PSS can compensate the inertia time delay of the excitation control system by extracting the speed deviation signal of the generator and compensating the inertia time delay of the excitation control system, so that the stabilizer can get the appropriate phase compensation and the speed deviation of generator is eliminated by integral loop.

Generator rotor kinetic energy:

$$K = \frac{1}{2} M k^2 \propto \frac{(P_m - P_e)^2}{T_M} \tag{4}$$

Generator rotor acceleration:

$$a_i = \frac{P_{mi} - P_{ei}}{M_i} \tag{5}$$

Therefore, $K$ and $a_i$ describe the rotational speed deviation of the generator when the power system is disturbed. The following feature sets can be selected: the Formula (4) is shown as the change value of the rotor speed per unit time, that is the rotor acceleration (the maximum, minimum, and average value of the rotor acceleration), and Formula (5) is shown as the rotor motion of the generator. It can take the difference between the maximum and minimum values of the kinetic energy of the generator rotor and the average kinetic energy of the generator rotor.

The construction of raw input features is a critical task for on-line evaluation of low frequency oscillation stability. Therefore, the construction of the original input features fundamentally determines the accuracy of online evaluation of low frequency oscillation stability. Through the analysis of the stability features of low frequency oscillation, the original input features which can fully reflect the change of the stability and dynamic features of low frequency oscillation at a certain time are complete. At the same time, in order to reflect the dynamic process of low frequency oscillation stability at different time, the original feature sets of disturbance occurrence time, disturbance end time, and different time of dynamic process are selected and the original input features of 4 and 5 typical moments are constructed. The 15-dimensional original input feature description at each moment is shown in Table 1.

**Table 1.** Original input features.

| Sequence Number | Feature Description |
| --- | --- |
| 1 | Total system load level |
| 2 | Mean value of mechanical power of generator |
| 3 | Maximum acceleration of generator rotor |
| 4 | Minimum value of generator rotor acceleration |
| 5 | The average value of the acceleration of the generator rotor |
| 6 | The maximum of the generator to accelerate the electromagnetic power |
| 7 | The minimum value of the generator to accelerate the electromagnetic power |
| 8 | The average value of the generator to accelerate the electromagnetic power |
| 9 | The difference between the maximum and minimum of the angular velocity of a generator |
| 10 | The difference between the maximum and minimum of the angular acceleration of generator |
| 11 | The difference between the maximum and minimum kinetic energy of generator rotor |
| 12 | Maximum kinetic energy of all generator rotors |
| 13 | Maximum value of generator excitation voltage per unit time |
| 14 | Minimum value of generator excitation voltage unit time variation |
| 15 | Average value of generator excitation voltage per unit time |

## 3. The Principle of the Improved XGboost Algorithm

### 3.1. The Principle of Wavelet Threshold De-Noising for Random Response Data

The random response data is the long-term dynamic response data in the daily operation of the power system, and the disturbance form and the specific occurrence position of the disturbance source can be ignored. The use of random response data for low frequency oscillation stability determination has the following two advantages:

1.  It can determine the low-frequency oscillation stability of the system through the machine learning method only by relying on the daily operation measurement data, avoiding the complicated construction process of the high-dimensional model and the error of the identification result caused by the difference between the model and the actual system.
2.  The electromechanical oscillation characteristic parameter identification process based on random response data does not need to prepare the disturbance experimental scheme in advance, and the system can be carried out under normal operating conditions, thereby overcoming the timeliness and credibility of evaluation method. The random response data provides real-time dynamic stability change information of the power system, which is suitable for online applications.

The random response data of power system collected by WAMS (wide area measurement system) can be expressed as:

$$y(n) = x(n) + v(n). \tag{6}$$

In the Formula (6), $y(n)$ is a signal containing noise; $x(n)$ is observed signal; $v(n)$ is Gauss white noise.

The key problem of wavelet threshold de-noising algorithm [28] is the selection of threshold and threshold function. The threshold method is as follows:

$$S = \sigma \sqrt{2 \ln N} \tag{7}$$

$$\sigma = \left( median \left| \omega_{j,k} \right| \right) / 0.6745. \tag{8}$$

In the Formula (7), $\sigma$ is the noise intensity, it is also the standard deviation of the noise signal; $N$ is the length of the signal. In the Formula (8), $median \left| \omega_{j,k} \right|$ is the median of wavelet coefficients on scale $j$.

The wavelet threshold method is applied to de-noise the random response data collected by WAMS.

### 3.2. The Principle of XGboost

XGboost is the abbreviation of extreme gradient rise, and it is a large-scale parallel algorithm. The XGboost model can be expressed as:

$$\hat{y}_i = \sum_{k=1}^{K} f_k(x_i), f_k \in F. \tag{9}$$

In the Formula (9), $i = 1, 2, \cdots, n$, $n$ is the number of samples; $F$ is a set that corresponds to all the regression trees, and $f_k$ is a function in $F$. When establishing a model, the best parameters should be selected [24] to make the target function minimum. The general objective function contains two items: the error term $L(\theta)$ (Error function) and the regularization term $\Omega(\theta)$ (Measuring the complexity of the model). The target function $f_{obj}^{(t)}$ is expressed as:

$$f_{obg}^{(t)} \approx \sum_{i=1}^{n} \left[ \begin{array}{c} l\left(y_i, \hat{y}_i^{(t-1)}\right) + g_i f_t(x_i) + \\ \frac{1}{2} h_i f_i^2(x_i) + \Omega(f_t) + C \end{array} \right] \tag{10}$$

In Formula (10), $g_i = \partial_{\hat{y}^{(t-1)}} l\left(y_i, \hat{y}_i^{(t-1)}\right)$; $h_i = \partial g_i$.

From Formula (10), objective function depends only on the first-order derivative and the second derivative of each data point on the error function.

Then the model complexity in the target function is defined. To refine $f$, the regression tree can be divided into the structural part of the tree $q$ and the weight part of the leaf $\omega$. That is:

$$\begin{array}{c} f_i(x) = \omega_{q(x)}, \ \omega \in R^T, \ q : R^T \to \{1, 2, \cdots T\} \\ f_i(x) = \omega_{q(x)}, \ \omega \in R^T, \ q : R^T \to \{1, 2, \cdots T\}. \end{array} \tag{11}$$

The number of leaf nodes is L1 regular, with a coefficient of $\gamma$, and the weight of leaves is L2 regular, with a coefficient of $\lambda$. The above two items are used to control tree growth to avoid overfitting to a certain extent. That is:

$$\Omega(f_t) = \gamma T + \frac{1}{2} \lambda \sum_{j=1}^{T} \omega_j^2. \tag{12}$$

Through Formula (12), the objective function seeks the maximum $\omega$ and the maximum gain of the corresponding function, and it transforms the problem into the minimum value problem of solving the quadratic functions. Solved:

$$\omega_j^* = \frac{-G_j}{H_j + \lambda}, \ f_{obj} = -\frac{1}{2} \sum_{j=1}^{T} \frac{G_j}{H_j + \lambda} + \gamma T. \tag{13}$$

In Formula (13), $f_{obj}$ is the scoring function of the evaluation model. If the value of $f_{obj}$ is smaller, the model is better. XGboost uses the "greedy method" to make $f_{obj}$ find the best tree structure, which is to add a new partition to the existing leaves each time and calculate the maximum gain that is obtained. Gain calculation formula is as follows:

$$f_{Gain} = \frac{1}{2} \left[ \frac{G_L^2}{H_L^2 + \lambda} + \frac{G_R^2}{H_R^2 + \lambda} - \frac{(G_L + G_R)^2}{H_L^2 H_R^2 + \lambda} \right] - \gamma. \tag{14}$$

In Formula (14), the first item represents the gain generated by the left subtree after the segmentation. The second item represents the gain generated by the right subtree after the segmentation. The third item represents the gain that does not carry out the segmentation. $\gamma$ represents the complexity cost of the new leaves due to the segmentation.

In this paper, the essence of the XGboost method is to parallel the Boosted Tree on a single CPU computer to improve the prediction accuracy of Boosted Tree.

### 3.3. Normalization Based on XGboost Features

If the data is not normalized, the loss function in XGboost can only choose linearity, which leads to the poor effect of the model. Therefore, Z-score normalization method [29] is adopted to normalize the original features.

The original feature set is $Y \in R^{n \times m}$ $Y \in R^{n \times m}$, where $n$ is the sample number and $m$ is the number of observed variables. The standardization of the original feature set $Y$ by z-score method is as follows:

$$y'_i = \frac{y_i - m(Y)}{s(Y)}. \tag{15}$$

In Formula (15), $y_i$ is the first sample, $m(Y)$ is the mean vectors of all values of the original feature set $Y$, $s(Y)$ is the standard deviation vectors of all values of the original feature set $Y$, and $y'_i$ is the sample data normalized by the sample $Y$.

Since z-score standardization uses the mean and variance of the entire data, the mean and variance of the data with different operation modes and different small interferences vary greatly. In order to adapt to the low frequency oscillation, the local data mean and variance are standardized.

The main idea of the local nearest neighbor standardization method is to standardize the mean and variance of the local neighbor samples consisting of $k$ nearest neighbors of sample $y'_i$. The formula is as follows:

$$y'_i = \frac{y_i - m(N_k(y_i))}{s(N_k(y_i))}. \tag{16}$$

In Formula (16), $k$ is the selected number of nearest neighbors, and $k$ must satisfy $k < n$, $N_k(y_i)$ is the data set of $k$ nearest neighbors determined by the Euclidean distance of sample $y_i$ in the original feature set $Y$. And $y_i^k$ is the $k$ nearest neighbor sample $y_i$, $d(y_i, y_j)$ is the Euclidean distance between two samples $y_i$ and $y_j$, then the relationship between $k$ nearest neighbor samples in $N_k(y_i)$ and the common of data set $N_k(y_i)$. The formulas are as follows:

$$d\left(y_i, y_i^1\right) < d\left(y_i, y_i^2\right) < \cdots < d\left(y_i, y_i^k\right) \tag{17}$$

$$N_k(y_i) = \left\{y_i^1, y_i^2, \cdots y_i^k\right\}. \tag{18}$$

## 4. Online Evaluation Model of Low Frequency Oscillation Stability Based on Improved XGboost

### 4.1. A Machine Learning Model for Low Frequency Oscillation stability Evaluation

According to the description of machine learning in [24], the problem of low frequency oscillation stability evaluation can be summarized to "machine learning that generalizes a specific problem model from limited observation" and "data analysis of various relationships implied in data from limited observation", while low frequency oscillation stability online evaluation belongs to the pattern recognition. The improved XGboost builds models with random response data and evaluates power system stability. Therefore, a machine learning model for low frequency oscillation stability evaluation can be constructed as shown in Figure 1.

The concrete steps are as follows:

1.  Set up simulation conditions to produce the random response data including various operation modes of the system, such as load fluctuation, switching and combination of generators, changing transformer tap, PSS parameters, and other factors.
2.  From the random disturbance data, select the characteristic variables that can characterize the system's health to form the input feature vector $x$ of the model.

3. The method of eigenvalue analysis is used to calculate the low frequency oscillation stability, and the stability of the system is judged by the criterion of damping ratio threshold. The stability can be expressed by the variable $y$ ($-1$ indicates the power system is unstable; 0 indicates that the long-time oscillation will cause harm to the system because of the weak damping; 1 indicates the power system is stable).

4. The original input features are de-noised by wavelet thresholding. A sample set $\{(x_1, y_1), \cdots, (x_n, y_n)\}$ of size $n$ is built and normalized, then $k$ samples are selected to train the model, and the indicator function is obtained to minimize the probability of classification error. Finally, the remaining $n-k$ samples are used to test the capability of improved XGboost machine learning model.

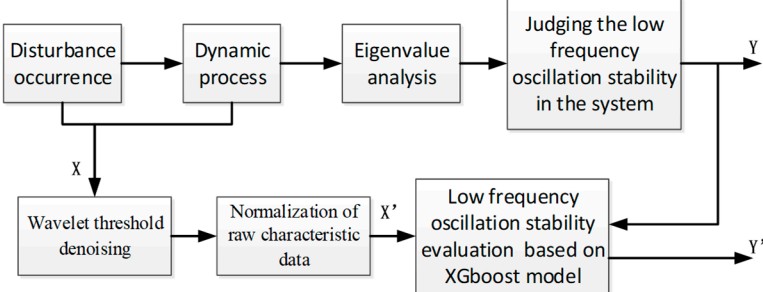

**Figure 1.** A machine learning model for low frequency oscillation stability evaluation

*4.2. Online Evaluation Process of Low Frequency Oscillation Stability Based on Improved XGboost*

Based on improved XGboost in this paper, the online evaluation framework of low frequency oscillation stability is proposed by two parts: offline training and online application, as shown in Figure 2. (Supplementary Materials are attached below the article.) The specific steps of the low frequency oscillation stabilization online evaluation are as follows:

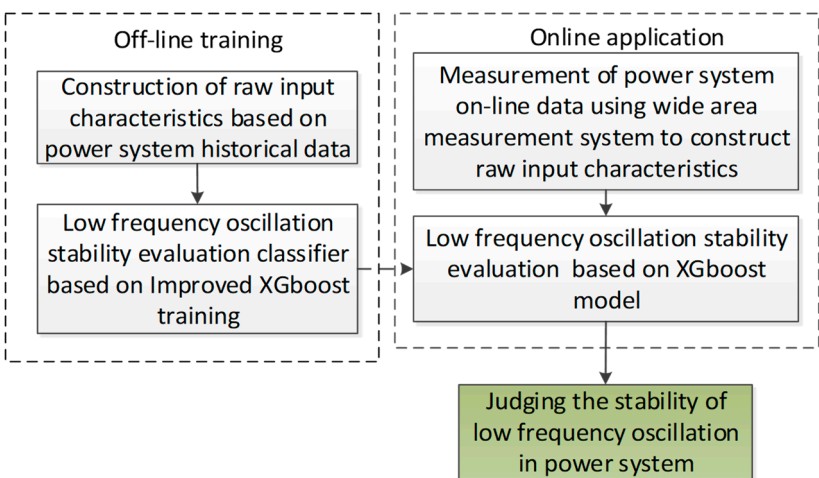

**Figure 2.** Process of power system low frequency oscillation stability evaluation.

Off-line training:

1. By setting up disturbances to simulate multiple operating conditions and operating scenarios of the power system, such as setting power fluctuations of different loads, switching combinations of different generators, and changing transformer taps, the original input features are de-noised using wavelet thresholds and normalized to construct a training set.

2.  The improved XGBoost algorithm is used to classify the original features to obtain an online evaluation model of low frequency oscillation stability. The classification value is obtained by analyzing the characteristic value of the operation mode.

Online application:

1.  De-noise and normalize the random response data obtained in the WAMS to construct the original input features.
2.  The original input characteristics are input into the low-frequency oscillation stability online evaluation model obtained through training to judge the low-frequency oscillation stability of the power system.

*4.3. Model Performance Evaluation*

The core issue in power system stability evaluation is to study which evaluation model is the most effective and how to evaluate the superiority of the model. In order to ensure the dynamic performance of the system in the actual system, the damping ratio should not be less than the threshold. In order to better reflect the correctness of the low-frequency oscillation stability evaluation results, the following indicators are used to define the model accuracy in the low-frequency oscillation stability evaluation:

$$a_{AMC} = \frac{f_{22} + f_{11} + f_{00}}{f_{22} + f_{11} + f_{00} + f_{21} + f_{12} + f_{20} + f_{20} + f_{10} + f_{01}} \tag{19}$$

$$a_{AMC} = \frac{f_{01} + f_{02}}{f_{22} + f_{11} + f_{00} + f_{21} + f_{12} + f_{20} + f_{20} + f_{10} + f_{01}} \tag{20}$$

$$a_{AMC} = \frac{f_{12} + f_{10}}{f_{22} + f_{11} + f_{00} + f_{21} + f_{12} + f_{20} + f_{20} + f_{10} + f_{01}} \tag{21}$$

$$a_{AMC} = \frac{f_{21} + f_{20}}{f_{22} + f_{11} + f_{00} + f_{21} + f_{12} + f_{20} + f_{20} + f_{10} + f_{01}} \tag{22}$$

$a_{AMC}$ is the proportion of the correct classification to the total classifications; $a_{FD}$ is the proportion of the unstable operating points identified as stable operating points to the total classifications; $a_{FM}$ is the proportion of the poor stable operating points identified as stable operating points to the total classifications, and $a_{FA}$ is the proportion of the stable operating points identified as unstable operating points to the total classifications. $f_{22}$ is the number correctly identified by the model when the electromechanical oscillation mode is strongly damped. $f_{11}$ is the number correctly identified by the model when the electromechanical oscillation mode is weakly damped. $f_{00}$ is the number correctly identified by the model when the electromechanical oscillation mode is negative damping. $f_{02}$ is that the mode of electromechanical oscillation is recognized as the number of strong damping when it is negative damping. $f_{01}$ is that the mode of electromechanical oscillation is recognized as the number of weak damping when it is negative damping. $f_{12}$ is that the mode of electromechanical oscillation is recognized as the number of strong damping when it is weakly damped. $f_{10}$ is that the mode of electromechanical oscillation is recognized as the number of negative damping when it is weakly damped. $f_{21}$ is that the mode of electromechanical oscillation is recognized as the number of weak damping when it is strongly damped. $f_{20}$ is that the mode of electromechanical oscillation is recognized as the number of negative damping when it is strongly damped.

The evaluation model can comprehensively evaluate the superiority of the model. The four evaluation indexes of $a_{AMC}$, $a_{FD}$, $a_{FM}$, and $a_{FA}$ fully reflect the correctness of the evaluation and the probability of each misjudgments in the evaluation, and they show the superiority of the low frequency oscillation stability.

## 5. Simulation Analysis

### 5.1. Example System

The eight-machine 36-node system shown in Figure 3 are selected as the test grid. The data set is simulated by MATLAB and its power system analysis toolbox (PSAT) for transient stability calculation and small signal stability calculation. To obtain the random response data in the power system, the operation state of the power system includes $0.86P_N, 0.88P_N, 0.90P_N, \cdots, 1.14P_N$, and so on, a total of 15 kinds of load levels (of which the generator changes according to the load level). In the 15 operation modes, the power flow calculation is carried out, the PSS parameters are changed, and the low frequency oscillation instability caused by the small disturbance is simulated. (The data is attached below the article.) Small disturbances are set as follows:

1.  The load fluctuation simulation small disturbance occurs on nine loads. The simulation setting is as follows: setting load fluctuation, the occurrence time is 0.9 s, and the end time is 1.1 s.
2.  Set part of the machine on eight generators to simulate small disturbance. The simulation setting is as follows: setting the cutting machine unit and its proportion, the time is 1 s, the time of excision is 1.1 s.
3.  Change the tap of transformer separately to simulate the occurrence of small disturbance. The occurrence time is 0.9 s.

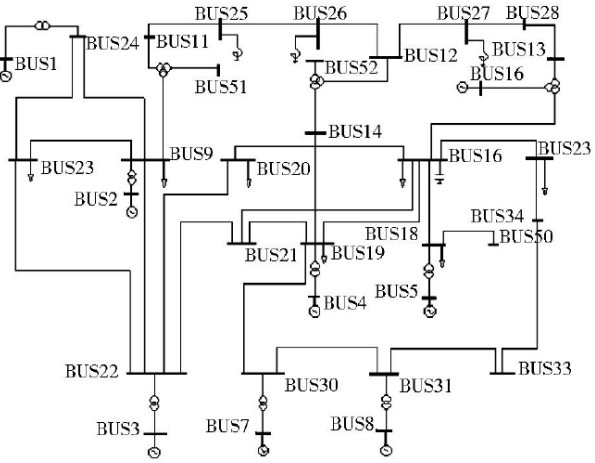

**Figure 3.** Eight-machine 36-node system structure.

At the end of the simulation, according to the eigenvalue analysis, if the damping ratio of all electromechanical oscillation mode is more than the threshold value, it is judged as stable (judged 1). if the damping ratio of any electromechanical oscillation mode is less than the threshold (0.03, 0.04, or 0.05) and more than 0, it is judged to be harmful to the system because of the long-time oscillation (judged 0). The damping ratio is less than 0, then it is judged to be low frequency oscillation and the system is unstable (judged −1). In this paper, 36,000 samples (13,500 load fluctuation samples, 12,000 cutting machine samples, 10,500 changing transformer taps samples) are obtained, of which 25,200 samples are used as training sets, and 10,800 samples are used as test sets. The training set is input into the model for training, and the test set is used to verify the validity of the model. (Negative sample ratio is 43%)

### 5.2. Optimal Original Input Feature Selection

By using the improved XGboost to evaluate the experiment, the best original input features of the model are obtained. Suppose $t_0$ is the time of small disturbance occurrence; $t_c$ is the small disturbance clearing time; $t_{c+i}$ is the $i$ cycle time after the small disturbance clearing time $t_c$. The features of $t_0$, $t_c$ and $t_{c+i}$ moments coincide with the physical quantities represented by characteristic 2~16 in Table 1.

Select 0.03 as the damping ratio threshold, and the selection results of different original features show that:

1.    All the original input features have fast calculation, and the calculation speed is within 0.012ms, which can meet the requirements of online application and have real time evaluation.
2.    The original input features are able to accurately assess the stability of low frequency oscillations. The correct rate of selecting the appropriate original feature input will reach 99.73%.
3.    Choosing long time scale can increase the accuracy of judgment, but it cannot meet the requirement of timeliness because of the need to collect the data of the long time, and timeliness and accuracy need to be judged comprehensively.
4.    By contrast, the best correct rate of the model results at the interval of three circumferential waves is 99.42%, and the requirement of meeting the time speed is higher than that of the choice interval of 4 weeks and two cycles, although it is lower than the correct rate of the interval of 30 circumferential waves, but the time of acquisition is 1/10.
5.    The addition of the original input features of the long-term scale has an interference effect on the training, and has no effect on improving the correct rate.

The comparison between Tables 2 and 3 shows that the original feature input of the original feature set $t_c, t_{c+3}, t_{c+6}, t_{c+9}$ can be selected in this paper to have highly effective accuracy and rapidity.

**Table 2.** Low frequency oscillation stability evaluation results with for feature selection at different times.

| Original Feature ($\xi$ = 0.03) | Correct Rate of Training | Single Sample Time Consuming | $a_{AMC}$ | $a_{FD}$ | $a_{FM}$ | $a_{FA}$ |
|---|---|---|---|---|---|---|
| $t_c, t_{c+3}, t_{c+6}, t_{c+9}$ | 99.99% | 0.012 ms | 99.42% | 0.22% | 0.25% | 0.11% |
| $t_0, t_c, t_{c+3}, t_{c+6}, t_{c+9}$ | 99.98% | 0.024 ms | 99.38% | 0.27% | 0.23% | 0.12% |

**Table 3.** Low frequency oscillation stability evaluation results with different interval cycle time.

| Original Feature ($\xi$ = 0.03) | Correct Rate of Training | Single Sample Time Consuming | $a_{AMC}$ | $a_{FD}$ | $a_{FM}$ | $a_{FA}$ |
|---|---|---|---|---|---|---|
| $t_c, t_{c+2}, t_{c+4}, t_{c+6}$ | 99.96% | 0.021 ms | 99.05% | 0.39% | 0.35% | 0.21% |
| $t_c, t_{c+3}, t_{c+6}, t_{c+9}$ | 99.99% | 0.012 ms | 99.42% | 0.22% | 0.25% | 0.11% |
| $t_c, t_{c+4}, t_{c+8}, t_{c+12}$ | 99.94% | 0.016 ms | 99.22% | 0.37% | 0.30% | 0.11% |
| $t_c, t_{c+10}, t_{c+20}, t_{c+30}$ | 99.98% | 0.017 ms | 99.31% | 0.32% | 0.22% | 0.15% |
| $t_c, t_{c+20}, t_{c+40}, t_{c+60}$ | 99.99% | 0.018 ms | 99.37% | 0.28% | 0.20% | 0.15% |
| $t_c, t_{c+30}, t_{c+60}, t_{c+90}$ | 99.99% | 0.019 ms | 99.73% | 0.08% | 0.13% | 0.06% |

The theoretical analysis shows that the characteristics of the three interval cycles can be used to characterize the change of the damping ratio of the system at that time, so it has a higher accuracy. However, the feature set of 30 interval-period times is selected to evaluate the high accuracy rate. Because the system low-frequency oscillation is stable and the sample data gap is larger, the characteristics of low-frequency oscillation stability are more obvious, but the rapidity of evaluation cannot be reflected.

The validity of the evaluation method is proved. After selecting three special operating states, the active oscillation curves of the generators after small disturbances are shown in Figures 4–6 by selecting the 0.03 threshold as an example. The results of machine learning evaluation are as follows: Figure 4 evaluation result is 1; Figure 5 evaluation result is 0; and Figure 6 evaluation result is −1.

As shown in Figures 4–6, the low frequency oscillation stability of the generator active power oscillation curve system is consistent with the on-line evaluation results, which further proves the effectiveness of the method.

In order to verify the relationship between the results and the eigenvalues, the topological structure is analysed. The results of eigenvalues analysis are shown in Tables 4–6 and the results of low

frequency oscillation stability are shown in Figures 4–6. In order to verify the relationship between the results and the eigenvalues, the topological structure is analyzed. The online evaluation results are consistent with the results of eigenvalue analysis, which proves that the online evaluation results of low-frequency oscillation stability are correct.

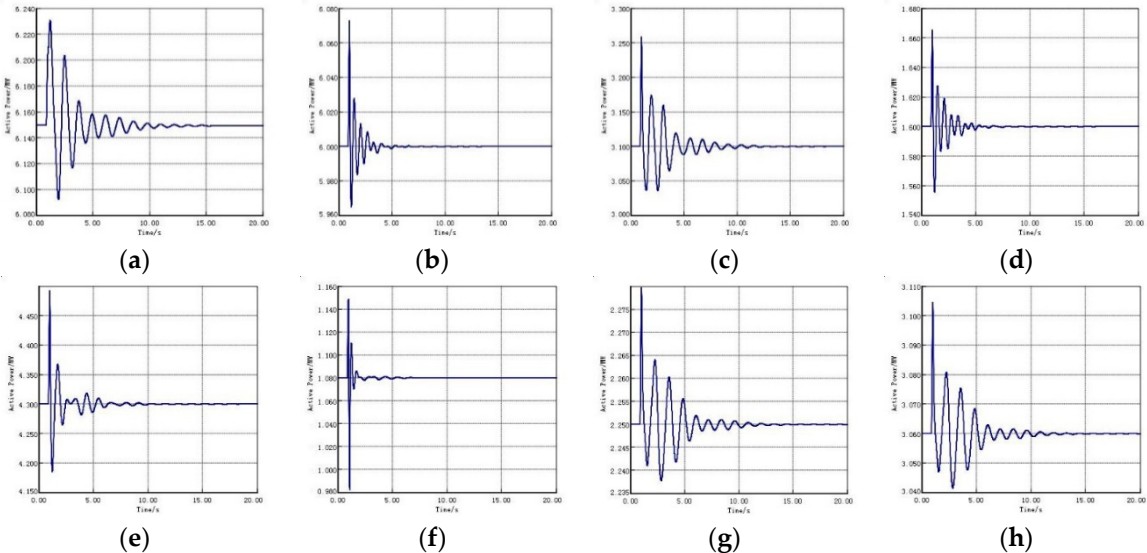

**Figure 4.** Generator power oscillation curve (1). (**a**) Generator 1; (**b**) Generator 2; (**c**) Generator 3; (**d**) Generator 4; (**e**) Generator 5; (**f**) Generator 6; (**g**) Generator 7; (**h**) Generator 8.

**Table 4.** Figure 4 corresponding topological eigenvalue analysis.

| Oscillation Modes | Eigenvalues | Damping Ratios (%) | Frequency |
|---|---|---|---|
| 1 | $-5.6431 \pm 13.1522i$ | 39.4300 | 2.1363 |
| 2 | $-0.8060 \pm 11.4808i$ | 7.0033 | 1.8272 |
| 3 | $-0.9118 \pm 10.4086i$ | 8.7263 | 1.6566 |
| 4 | $-0.8552 \pm 7.5235i$ | 11.2942 | 1.1974 |
| 5 | $-0.7991 \pm 7.0328i$ | 11.2895 | 1.1193 |
| 6 | $-0.3522 \pm 5.6326i$ | 6.2412 | 0.8965 |
| 7 | $-0.1819 \pm 4.6283i$ | 3.9025 | 0.7366 |

**Figure 5.** Generator power oscillation curve (0). (**a**) Generator 1; (**b**) Generator 2; (**c**) Generator 3; (**d**) Generator 4; (**e**) Generator 5; (**f**) Generator 6; (**g**) Generator 7; (**h**) Generator 8.

**Table 5.** Figure 5 corresponding topological eigenvalue analysis.

| Oscillation Modes | Eigenvalues | Damping Ratios (%) | Frequency |
|---|---|---|---|
| 1 | $-5.2415 \pm 14.2910i$ | 34.4339 | 2.2745 |
| 2 | $-0.7205 \pm 10.6761i$ | 6.7331 | 1.6992 |
| 3 | $-0.8201 \pm 9.703i$ | 8.4214 | 1.5443 |
| 4 | $-0.6480 \pm 7.6864i$ | 8.4003 | 1.2233 |
| 5 | $-0.5760 \pm 6.5670i$ | 8.7377 | 1.0452 |
| 6 | $-0.2034 \pm 5.4348i$ | 3.7398 | 0.8650 |
| 7 | $-0.0703 \pm 4.3686i$ | 1.6088 | 0.6953 |

**Figure 6.** Generator power oscillation curve ($-1$). (**a**) Generator 1; (**b**) Generator 2; (**c**) Generator 3; (**d**) Generator 4; (**e**) Generator 5; (**f**) Generator 6; (**g**) Generator 7; (**h**) Generator 8.

**Table 6.** Figure 6 corresponding topological eigenvalue analysis.

| Oscillation Modes | Eigenvalues | Damping Ratios (%) | Frequency |
|---|---|---|---|
| 1 | $-7.5168 \pm 13.3203i$ | 49.1460 | 2.1200 |
| 2 | $-0.6974 \pm 10.3277i$ | 6.7376 | 1.6437 |
| 3 | $-0.7798 \pm 9.4618i$ | 8.2137 | 1.5059 |
| 4 | $-0.5609 \pm 7.7314i$ | 7.2354 | 1.2305 |
| 5 | $-0.5191 \pm 6.4462i$ | 8.0262 | 1.0259 |
| 6 | $-0.1242 \pm 5.3306i$ | 2.3284 | 0.8484 |
| 7 | $0.0281 \pm 4.2407i$ | -0.6630 | 0.6749 |

*5.3. Evaluation Performance of Models with Different Samples and Different Damping Ratios*

The original feature set with different samples and the original feature set with different damping ratio thresholds are evaluated by improved XGboost. The results are shown in Table 7. The performance of model evaluation is analyzed from the following two perspectives:

1. Performance comparison of the original feature set samples with the damping ratio of 0.03, 0.04, and 0.05 is carried out.
2. The model performance analysis is carried out for the original feature set samples considering the load fluctuation, the generator switching, and the transformer tap, and the model performance analysis of the original feature set samples was carried out based on the first three small disturbance cases.

The results of different sample and different damping ratio threshold show that:

1.  The random response data contains more perturbation types, and the accuracy of the evaluation is higher.
2.  The selection damping ratio threshold is 0.03, and the evaluation model has the highest evaluation accuracy.

**Table 7.** Low frequency oscillation stability evaluation results with different samples and different damping ratios thresholds.

| Sample Selection | Damping Ratio Threshold | $a_{AMC}$ | $a_{FD}$ | $a_{FM}$ | $a_{FA}$ |
|---|---|---|---|---|---|
| Load fluctuation | 0.03 | 98.55% | 0.61% | 0.43% | 0.41% |
| | 0.04 | 98.29% | 0.76% | 0.53% | 0.42% |
| | 0.05 | 98.31% | 0.77% | 0.44% | 0.48% |
| Generator input and excision occurrence | 0.03 | 98.65% | 0.51% | 0.45% | 0.39% |
| | 0.04 | 98.55% | 0.57% | 0.48% | 0.40% |
| | 0.05 | 98.34% | 0.59% | 0.50% | 0.57% |
| Change the hair generation time of transformer | 0.03 | 98.42% | 0.59% | 0.55% | 0.44% |
| | 0.04 | 98.30% | 0.64% | 0.55% | 0.51% |
| | 0.05 | 98.19% | 0.65% | 0.51% | 0.65% |
| Small disturbance occurrence | 0.03 | 99.42% | 0.22% | 0.25% | 0.11% |
| | 0.04 | 99.12% | 0.33% | 0.28% | 0.27% |
| | 0.05 | 98.92% | 0.37% | 0.35% | 0.36% |

*5.4. Online Evaluation Results of Low Frequency Oscillation Stability in Different Models*

The traditional SSI algorithm uses the singular value decomposition method, while the prony algorithm uses the method of quasi-sum sampling data. The calculation speed is slow in seconds, and its noise resistance is poor, and the shortcoming of real-time performance cannot be guaranteed.

To embody the accuracy and real-time performance of machine learning algorithm in low frequency oscillation stability evaluation, as a contrast, the SVM, the random forest, XGboost, and the improved XGboost algorithm proposed in this paper are used to carry out the comparison test of low frequency oscillation stability evaluation.

The results in Table 8 of the same model selection shows that:

1.  Compared with SVM, XGboost, and random forest, the improved XGboost has better accuracy and rapidity.
2.  The improved XGboost has the highest evaluation accuracy in reducing the error evaluation rate of unstable samples, and it can better prevent unstable samples from being recognized as stable samples so that it cannot be alarmed in time.

**Table 8.** Low frequency oscillation stability evaluation results with different models.

| Model ($\xi = 0.03$) | Single Sample Time Consuming | $a_{AMC}$ | $a_{FD}$ | $a_{FM}$ | $a_{FA}$ |
|---|---|---|---|---|---|
| SVM | 0.112 ms | 75.70% | 19.75% | 3.11% | 1.44% |
| Improved XGboost | 0.012 ms | 99.42% | 0.22% | 0.25% | 0.11% |
| XGboost | 0.067 ms | 94.79% | 2.50% | 1.48% | 1.23% |
| Random forest | 0.050 ms | 93.81% | 4.64% | 1.11% | 0.50% |

Therefore, improved XGboost algorithm has the features of high accuracy and real-time in the evaluation of low-frequency oscillation stability.

### 5.5. Model Evaluation Performance Considering Wide Area Measurement System Noise

The results of noiseless and noisy under different models indicate:

The Gauss white noise of 50dB, 30dB, and 10dB is added to the original data to simulate the measurement error of the wide area measurement system, and the accuracy rate comparison of the different model low frequency oscillation stability on-line evaluation tests is carried out. The results are shown in Table 9.

**Table 9.** Low frequency oscillation stability evaluation based on noise signal model.

| Model (ξ = 0.03) | Accuracy Rate of Low Frequency Oscillation Stability Evaluation | | | |
| --- | --- | --- | --- | --- |
| | Noiseless | 50db | 30db | 10db |
| Improved XGboost | 99.42% | 99.31% | 98.72% | 97.24% |
| XGboost | 94.79% | 91.88% | 89.62% | 85.41% |
| Random forest | 93.81% | 90.66% | 86.06% | 81.19% |

Under the same signal to noise ratio, the improved XGboost has the highest evaluation accuracy, and improved XGboost is slightly better than XGboost and random forest in anti-noise. It shows that improved XGboost can play the role of noise filtering, and all the two have strong generalization ability.

### 5.6. Actual System Simulation Analysis

The selection of Hebei southern power grid as the actual test system is shown in Figure 7. The data set is simulated by MATLAB. To obtain the random response data in the power system, multiple actual run modes are selected. In the operation modes, the power flow calculation is carried out, the PSS parameters are changed, and the low frequency oscillation instability caused by the small disturbance is simulated. Small disturbances are set as follows: (1) The load fluctuation simulation small disturbance occurs; (2) Set part of the machine on generators to simulate small disturbance; and (3) Change the tap of transformer separately to simulate the occurrence of small disturbance.

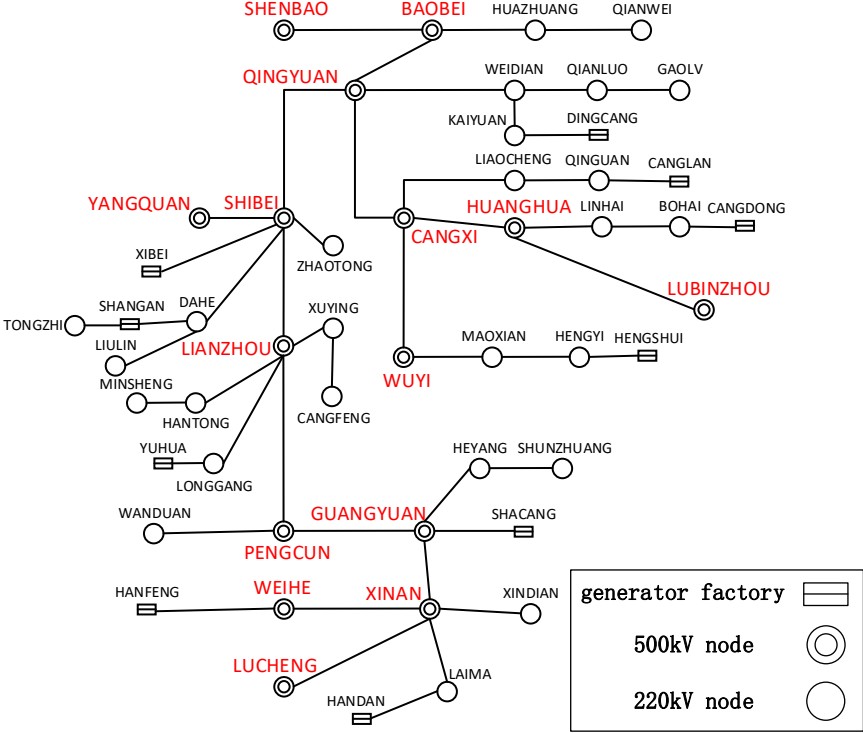

**Figure 7.** Hebei southern power grid structure.

In this paper, 24,000 samples (12,000 load fluctuation samples, 6000 cutting machine samples, 6000 changing transformer taps samples) are obtained, of which 16,800 samples are used as training sets and 7200 samples are used as test sets. The training set is input into the model for training, and the test set is used to verify the validity of the model. (Negative sample ratio is 21%)

To sum up, the results Tables 10 and 11 show that in a real environment the online evaluation method of low frequency oscillation stability based on improved XGboost algorithm also has the features of high accuracy of evaluation, efficiency of calculation, strong anti-noise signal ability, and low error rate of unstable sample evaluation.

**Table 10.** Low frequency oscillation stability evaluation in Hebei southern power grid.

| Model ($\xi$ = 0.03) | Single Sample Time Consuming | $a_{AMC}$ | $a_{FD}$ | $a_{FM}$ | $a_{FA}$ |
|---|---|---|---|---|---|
| Improved XGboost | 0.018 ms | 98.85% | 0.65% | 0.33% | 0.17% |

**Table 11.** Low frequency oscillation stability evaluation based on noise signal model.

| Model ($\xi$ = 0.03) | Accuracy Rate of Low Frequency Oscillation Stability Evaluation | | | |
|---|---|---|---|---|
| | Noiseless | 50db | 30db | 10db |
| Improved XGboost | 98.85% | 98.51% | 98.02% | 96.54% |

## 6. Conclusions

In this paper, an online evaluation method of power system low frequency oscillation stability based on improved XGboost algorithm is proposed. The simulation research on the eight-machine and 36-node test grids and Hebei southern power grid are carried out. (Supplementary Materials are attached below the article.) The conclusions are as follows:

1. The improved XGboost algorithm uses random response data to consider grid uncertainty and it can solve the problem of online evaluation of low-frequency oscillation stability. Meanwhile, it has the characteristics of high accuracy for evaluating the stability of low frequency oscillations. It can significantly reduce the misjudgment in actual power grid.
2. The proposed evaluation model of low frequency oscillation stability has high efficiency of calculation and it can be applied online. When the low frequency oscillation of the actual power grid runs fast alarm, the power grid staff can take emergency preventive measures before the harm happened to avoid the loss.

Improved XGboost model is widely used in data mining and artificial intelligence. The research shows that the improved XGboost model can be applied to the low frequency oscillation stability analysis. However, this study will not be able to urgently control the unstable power system to suppress low-frequency oscillations because it only performs stable online evaluation of low-frequency oscillations. So that become a problem for subsequent research.

**Supplementary Materials:** Source code and data are available here: http://www.mdpi.com/1996-1073/11/11/3238/s1.

**Author Contributions:** W.H. conceived the structure and research direction of the paper; J.L. wrote the paper and completed the simulation for case studies; Y.J. provided algorithms; F.W. wrote programs and analyzed the data. X.W. verified the method. E.C. collected the data and processed it.

**Funding:** This research received no external funding.

**Conflicts of Interest:** The authors declare no conflict of interest.

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
