# Peer review of "Online Evaluation Method for Low Frequency Oscillation Stability in a Power System Based on Improved XGboost"

_energies, doi:10.3390/en11113238_

Round 1

Reviewer 1 Report

Low-frequency oscillations are in general problem in weakly connected power systems or in weakly connected groups of generators. Soundness of research is high however the reader can misunderstand research conducted in this work together with conclusions. Authors properly stated in the conclusion that they did not conduct experimental research. The proposed algorithm has been tested in a simulation environment and not in a real environment. So the suggestion to the authors is to state this in the title of the work. Also, partially algorithm can be tested in the SMIB topology and authors are encouraged to do real testing of the algorithm.  

For simulation purpose and verification of the algorithm, eigenvalue analysis should be conducted for the simulated topology and results of the algorithm should be clearly associated with the eigenvalues responsible for low-frequency oscillations. Different cases should be simulated to get a clear picture of algorithm capabilities.

In low dumping and in negative dumping conditions simulation results should be presented, eigenvalue analysis presented and make a clear relationship with algorithm results. 

Overall the idea is good, the article is well structures however some extra effort should be placed to finish the article. Especially in the simulation part and in rewriting conclusion with new experimental results presented. 

Author Response

Thank you for pointing out these problems. See WORD for your comments.

Reviewer 2 Report

Dear authors, dear editor, 

This manuscript proposes a machine learning method for evaluating the low frequency oscillation power system stability. It is well written and developed. 

Specific comments follow below for each section. 

0. Introduction

Bibliography shall be more recent. It is expected that priority is given to articles published less than five years ago. In this manner, the authors are better able to establish the state-of-the-art on their subject and compare it with their work. 

Authors’ contribution is clear to the reviewer, however, the could further elaborate in explaining the benefits of their approach to other similar methods of the bibliography. 

Sections’ numbering usually starts from 1 rather 0. 

1. The construction of the original input feature

Would the authors improve their writings at this section, this manuscript’s readability could be enhanced. Also, part of the information provided here is known to the potential audience and hence can be reduced and/or omitted. 

2. The principle of the improved XGboost algorithm

2.1. The priniple of wavelet threshold de-noising for random response data, 2.2. The priniple of XGboost, and 3. Normalization based on XGboost features

This is a valid approach. 

3. Online evaluation model of low frequency oscillation stability based on improved XGboost

3.1. A machine learning model for low frequency oscillation stability evaluation

How the authors safeguard that the available/simulated data for machine learning are adequate? 

3.2. Online evaluation process of low frequency oscillation stability based on improved XGboost

On which degree, normal operations of the power system, such as the connection and the disconnection of system components could affect the prediction accuracy of this method? 

3.3. Model performance evaluation

As above, this is a valid approach. 

4. Simulation analysis

Source code and data shall be openly offered as supplementary material to this manuscript. This will improve its replicability and facilitate other researchers who wish to build upon its findings. 

4.1. Example system

Why this system has been chosen? 

4.2. Optimal original input feature selection, 4.3. Evaluation performance of models with different samples and different damping ratios, 4.4. Online evaluation results of low frequency oscillation stability in different models

Results show good consistency. 

4.5. Model evaluation performance considering wide area measurement system noise

It is of paramount importance that the authors are able to provide a comparative analysis of the proposed and other methods. 

5. Conclusion

Conclusions are supported by the analysis. 

Having mentioned the above, this manuscript is proposed to be published after minor revision. 

Sincerely yours, reviewer

Author Response

Thank you for pointing out these problems. See WORD for comments.

Round 2

Reviewer 1 Report

Thanks to the authors for implementing remarks into the article. 

Reviewer 2 Report

Dear authors, dear editor, 

Authors have adequately covered all reviewer’s comments. Henceforth, the reviewer strongly encourages the editor to publish this manuscript. 

Sincerely yours, reviewer